# STDACN: A Spatiotemporal Prediction Framework based on Dynamic and Adaptive Convolution Networks

## Abstract

With the rapid advancement of sensor technologies, analyzing and modeling large spatiotemporal datasets has become crucial, enabling system state predictions for intelligent transportation, urban planning, public safety, and environmental protection. Current models—statistical, classical deep learning (e.g., TCN, GCN), and large-scale methods—struggle with noise, complexity, high dimensionality, and dynamics, with static TCN/GCN structures limiting performance and large models facing high computational costs, keeping classical methods relevant. This paper proposes a spatiotemporal prediction framework based on dynamic and adaptive convolution networks (STDACN), which overcomes weight-sharing limits, featuring a high-order gated TCN with recursive causality to capture temporal dependencies and an adaptive GCN for spatial topologies, boosting efficiency and generalization. Excelling in traffic, weather, and population predictions across varied scales, STDACN offers a simple yet innovative path for classical deep learning in complex spatiotemporal modeling.

## 1 Introduction

Due to the growing availability and significance of large spatiotemporal datasets like maps, remote sensing images, population, and traffic data, spatiotemporal data mining and prediction Hamdi et al. (2022) has emerged as a key focus in smart cities and spatial big data, widely applied in weather, traffic flow, and earthquake forecasting Yuan et al. (2024). These models analyze time series relationships and capture spatiotemporal dependencies in graph-based spatial networks (e.g., traffic road networks), delivering valuable applications in intelligent transportation, urban planning, public safety, and environmental protection.

Currently, spatiotemporal data models Hamdi et al. (2022) include statistical models, classical deep learning models, rising large models Fang et al. (2024), etc. Real-world data, with its complex features, high dimensions, frequency, and noise, challenges predictive models. Despite large models' growing popularity, their high computational and inference demands sustain the development of classical spatiotemporal deep learning, led by Temporal Convolutional Networks (TCN) and Graph Convolutional Networks (GCN) for effective data modeling. For instance, Graph WaveNet Wu et al. (2019) utilizes dilation convolution to capture time-dependent features and multigraph diffusion convolution to extract spatial features. CSTN Song et al. (2020) utilizes TCN operations to capture temporal evolution context, local spatial context, and global correlation context. STHGCN Wang et al. (2022) leverages TCN Bai et al. (2018) and GCN Kipf & Welling (2017) operations to extract higher-order spatiotemporal dependencies from spatiotemporal data. TCGCN Wang et al. (2024a) integrates cross-dimensional attention to discern features and relationships across different dimensions of spatiotemporal data.

However, there is a lack of research on the spatiotemporal dynamic characteristics of these models, particularly given the recent surge in large model development. The application of simple methods from classical deep learning for dynamic adaptive learning of spatiotemporal information is uncommon, highlighting the limitations of traditional and effective spatiotemporal prediction models.

Fig. 1 illustrates that spatiotemporal data comprises three dimensions, with varying attributes over time. Convolution operators in deep learning algorithms exhibit translation invariance, facilitated by

local connectivity and shared weights. Studies indicate that strict weight sharing may not optimize feature extraction in spatiotemporal data analysis with distinct time, space, and feature dimensions. Utilizing shared convolution weights for multidimensional feature extraction along the time dimension is suboptimal in such scenarios. Conversely, a Temporal Adaptive Dynamic Adjacency Matrix-based approach, such as WAN, enables dynamic fusion of spatial information without a substantial parameter increase.

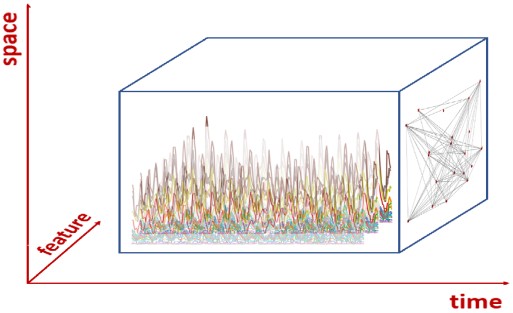

Figure 1: Tensor and Characteristics of ST Data

To address this issue, we propose a spatiotemporal prediction framework based on dynamic and adaptive convolution networks (STDACN), which overcomes the weight sharing limitations. It features a novel high-order dynamic gated TCN with a response time readout function to adjust temporal kernels and mitigate gradient vanishing, alongside a global adaptive GCN module using dynamic diffusion convolution to capture spatial topologies. This approach adapts to diverse temporal and spatial scales, excelling in analyzing complex, dynamic, high-dimensional spatiotemporal data. The key contributions of this study are summarized as follows:

- We introduce a novel spatiotemporal prediction framework, termed STDACN, which offers dynamic adaptive modeling capabilities to accommodate varying time intervals and spatial resolutions. This model, characterized by its simplicity and clarity, establishes a novel theoretical foundation and practical avenue for leveraging traditional deep learning approaches in the modeling of intricate spatiotemporal datasets.

- For dynamic and high-order time dependence, the high-order dynamic gating TCN is combined with a dynamic recursive causality mechanism to break the weight sharing constraint. By combining the adjacency matrix with a dynamic recursive causality mechanism, the time dependence of data is captured, and the gradient disappearance is effectively avoided.

- An adaptive dynamic graph convolutional network (GCN) utilizing diffusion convolution is developed to address spatial correlation. The conventional GCN convolution module is expanded to a dynamic GCN using a regular network. The dynamic diffusion convolution incorporates both global dynamic and temporal dynamic spatial relationship information.

- Project STDACN was evaluated using actual spatiotemporal data sets (including traffic data and electricity data) and showed a reduction in prediction error of approximately 4.7% - 1.2% compared to all baseline models.

## 2 METHOD

This section includes model structure, dynamic time convolution, adaptive GCN module, and decoding, and introduces the design details of STDACN.

### 2.1 OVERVIEW OF THE STDACN'S STRUCTURE

The whole framework of STDACN shown in Fig. 2 is composed of an encoder and a decoder, which respectively extract spatiotemporal dependent feature embedding and map this embedding into a prediction vector. This section will detail the core layers of STDACN based on the input $X_s$.

The encoder consists of multi-layer spatiotemporal modules: High-order gated, Dynamic CNN, and Dynamic GCN layers. The High-order gated and Dynamic CNN layers form the TCN for extracting

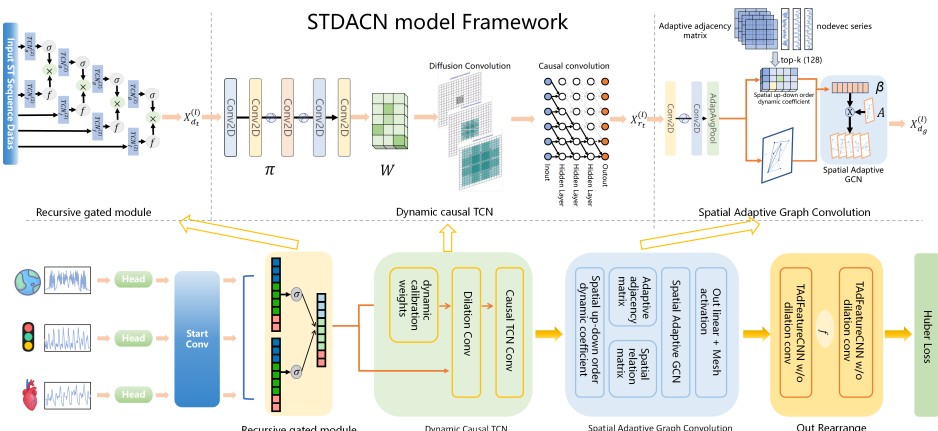

Figure 2: The STDACN structure includes an encoder and a decoder.

high-order dynamic temporal dependencies, while the Dynamic GCN layer establishes spatial topology dependence using adaptive relations and calibration vectors. The decoder, comprising a linear layer and two Dynamic CNN layers, translates the spatiotemporal embedding into a prediction vector. Following the approach of previous deep-learning models for spatiotemporal prediction Wang et al. (2022), STDACN initially conducts a 2D convolution on the time and space dimensions of input matrix $X \in \mathbb{R}^{F \times N \times P}$, where $F$ represents the convolution channel and $H$ denotes the hidden feature dimensions.

## 2.2 THE TEMPORAL CONVOLUTION LAYER TCN

Spatiotemporal data is essentially a multivariate time series, which has obvious temporal dependence. The efficient and dynamic features of temporal dependence are very important for spatiotemporal prediction. WaveNet Wu et al. (2019) uses dilated causal convolution and a gated mechanism as TCN to capture temporal trends, expanding the receptive field with layer depth. To address the long-term dependency issues of RNN methods Fan et al. (2022), transformer-based architectures Zhou et al. (2021) are widely adopted for temporal interaction, though dot-product self-attention is less effective. For efficiency, STDACN's TCN employs recursive gated convolution $g^n$ Conv, using dynamic kernels for high-order temporal interactions.

### 2.2.1 THE RECURSIVE GATED TEMPORAL CONVOLUTION

The calculation process of $g^n$ Conv is shown in Fig. 2 Sub-chart A. Assume that the input of the $l$-th layer is spatiotemporal data $X^{(l)} \in \mathbb{R}^{H \times N \times P^{(l)}}$, where $l \in \{0, \cdots, L-1\}$ and the $P^{(l)}$ is the temporal dimensions of the $l$-th layer, and then $g^n$ Conv has expressed as follows:

$$
\begin{aligned}
TCN_g^{(k)}(X_k^{(l)}) =& \sigma\left(\text{conv2D}_g(X_k^{(l)})\right), \\
TCN_f^{(k)}(X^{(l)}) =& f\left(\text{conv2D}_f(X^{(l)})\right), \\
X_{k+1}^{(l)} =& TCN_g^{(k)}(X_k^{(l)}) \odot TCN_f^{(k)}(X^{(l)})
\end{aligned}
\tag{1}
$$

where $\text{conv2D}_g$ and $\text{conv2D}_f$ are 2D convolution operations with kernel size $k = 1 \times 3$ and padding $p = 0 \times 1$, preserving the temporal dimension length and spatial order. The activation functions $\sigma(x) = \frac{1}{1+e^{-1}}$, $f(x) = x \tanh\left(\ln\left(1 + e^x\right)\right)$ Misra (2019). $K$ is the recursive order of $g^n$ Conv, $k = 0, 1, \cdots, K-1$.

Compared with the standard Gated TCN like in Wu et al. (2019), the activation function $\tanh$ is instead by the $\text{mish}$ function, because the range of $\text{mish}$ has no positive boundary, so it can avoid the disappearance of gradient generated in recursive transfer. It can be seen from the formula (1) that the $g^n$ Conv has stronger information filtering ability than the standard Gated TCN, because its output is $X_{k+1}^{(l)}$, and the filter has larger receptive field and higher interactive capacity of time information.

### 2.2.2 THE DYNAMIC CAUSAL TEMPORAL CONVOLUTION

Beyond requiring recursive high-order temporal dependence and dynamic time-based adjustments, the core of this section focuses on constructing the channel dimension $(X_K^{(l)})$ temporal filters for dynamic convolution to enhance prediction accuracy. The core calculation process is shown in Fig. 2 Sub-chart B, and formally, the process can be obtained by:

$$X_{dt}^{(l)} = \sigma \left( X_K^{(l)} \star (\pi.W) \right), \tag{2}$$

where $X_K^{(l)} \in \mathbb{R}^{H \times N \times P^{(l)}}$ is the input of the dynamic TCN and is the output of formula (1), $\star$ is the convolution operator with the kernel weights $W \in \mathbb{R}^{c_{out} \times c_{in} \times 1 \times k}$ and the dilation rate $r_d$, $\sigma = \text{mish}$ is the active function, and $\pi = \Pi \left( X_K^{(l)} \right) \in \mathbb{R}^{c_{in}}$ is the dynamic calibration weights which is the dynamic causal TCN's key module. It can be seen from formula (2) and Fig. 2 Sub-chart B that the calibration generation function $\pi = \Pi(X_K^{(l)}) \in \mathbb{R}^{c_{in}}$ could extract dynamic change information of spatiotemporal data along the input channel dimension $X_K^{(l)}$. We initially employ a down-up type convolution to transform the data for enabling $\Pi$ with learning capability, that is:

$$X_{ud}^{(l)} = \sigma \left( \text{conv2D}_r(\text{norm}(\sigma(\text{conv2D}_{\frac{1}{r}}(X_K^{(l)})))) \right) \tag{3}$$

where $\sigma$ is the activate function, $\text{conv2D}_{\frac{1}{r}}$ and $\text{conv2D}_r$ are the 2d convolutions which input channels are $\frac{c_{in}}{r}$ and $c_{in}$ and $r$ is a hyperparameter. The adaptive average pool function is used to extract features from space-time dimension of $X_{ud}^{(l)} \in \mathbb{R}^{H \times N \times P^{(l)}}$, which is formulated as:

$$X_{rt}^{(l)} = \text{mean}(X_{ud}^{(l)}) \in \mathbb{R}^{H \times 1 \times \frac{P^{(l)}}{r_t}}, \tag{4}$$

$$\pi = \Pi(X_K^{(l)}) = \text{fc}(X_{rt}^{(l)}) \tag{5}$$

where $r_t$ is the hyperparameter for temporal dimension average partitioning.

### 2.3 THE SPATIAL ADAPTIVE GRAPH CONVOLUTION

Spatiotemporal data represent multivariable time series with inherent spatial patterns. Enhancing the integration of spatial dynamics in spatiotemporal data analysis is crucial, building upon the dynamic examination of temporal characteristics. By combining pre-defined spatial dependencies and self-learned hidden graph dependencies, we proposed the following GCN operator:

$$Z = \sum_{k=0}^{K} P^k X W_{kP} + \tilde{A}_{apt}^k X W_{kA}, \tag{6}$$

$$\tilde{A}_{adp} = \text{SoftMax} \left( \text{ReLU} \left( E_1 E_2^T \right) \right) \tag{7}$$

where $K$ is the order of the GCN, $P$ is the pre-defined spatial structure, $\tilde{A}_{adp}$ is a self-adaptive adjacency matrix constructed by randomly initializing two learnable node embedding dictionaries $E_1, E_2 \in \mathbb{R}^{N \times d}$, with $W_{kP}$ and $W_{kA}$ as GCN parameters.

However, the spatial topology structures in formulas (6, 7) are global, that is, all samples at any time share $P$ and $\tilde{A}_{adp}$. DGCRN Li et al. (2021) uses dynamic topology with RNN, while increasing computation parameters and facing gradient disappearance issues. Based on the above factors, STDACN develops an adaptive dynamic GCN by incorporating a spatial network dynamic generation factor, denoted as $\beta$. This approach is inspired by the global topology generation dynamic formula(2). The module has shown in Fig. 2 Sub-chart C and could be formulated as follows:

$$X_{dg}^{(l)} = \sum_{k=0}^{K} (\beta P)^k X_{dt}^{(l)} W_{kP} + \left( \beta \tilde{A}_{apt} \right)^k X_{dt}^{(l)} W_{kA}, \tag{8}$$

$$\mathcal{B} = \text{mean} \left( \text{Conv2D}_r \left( \sigma \left( \text{Conv2D}_{1/r}(X_{dt}^{(l)}) \right) \right) \right) \tag{9}$$

where $X_{dt}^{(l)}$ is the input of dynamic GCN and the output of formula (2).It can be seen from formula (8) and Fig. 2 Sub-chart C that the adjacency matrices are $\beta P, \beta \tilde{A}_{apt} \in \mathbb{R}^{P^{(l)} \times N \times N}$ which dynamic change over time. Among them, calculating the spatial calibration weight $\beta = \mathcal{B}(X_{dt}^{(l)})$ is the key point, which is similar to formulas (3) and (4), its generation process can be described by formula (9). Finally, the output of layer $l$ is $X^{(l+1)} = \text{norm}(X_{dg}^{(l)} + X^{(l)})$, where norm normalizes the data to enhance training.

## 2.4 THE DECODER AND LOSS FUNCTION

After the encoder encodes spatiotemporal dependence into $X^{(L-1)} \in \mathbb{R}^{H \times N \times P^{(L-1)}}$, the STDACN decoder, featuring a linear layer and two dynamic TCN layers (as seen in Fig. 2), transforms the input temporal dimension $P^{(L-1)}$ to 1, encoding it into the output temporal feature dimension $H$. Subsequently, two-layer dynamic temporal convolutions convert this into the prediction temporal dimension. It can be a formula as:

$$X_f = \sigma \left( \text{liner}(X^{(L-1)}) \right) \in \mathbb{R}^{H \times N \times 1}, \tag{10}$$

$$\hat{Y} = f_{d1}(f_{d2}(X_f)), \tag{11}$$

where $f_{d1}, f_{d2}$ are dynamic convolutions like formula (2) without dilation, obtaining the final prediction $\hat{Y} \in \mathbb{R}^{N \times Q}$.

The STDACN model employs Huber loss Huber (1992) for its reduced sensitivity to outliers compared to squared error loss, where $Y$ represents the real training values, the formula has shown as follows:

$$L_{\text{Huber}}(\hat{Y}, Y) = \begin{cases} \frac{1}{2}(Y - \hat{Y})^2 & |Y - \hat{Y}| \leq \delta \\ \delta|Y - \hat{Y}| - \frac{1}{2}\delta^2 & \text{otherwise} \end{cases}, \tag{12}$$

## 3 EXPERIMENTAL

### 3.1 EXPERIMENTAL DESIGN

**Datasets and Baselines.**The STDACN model was assessed using four public datasets: METR-LA and PEMS-BAY for evaluating its robustness in handling missing data and ability to capture complex topologies and long-term dependencies, respectively, and PEMS03 and PEMS08 for assessing its performance in modeling sparse nodes and implicit spatial relationships. To comprehensively assess the efficacy, we conducted a comparative analysis with 12 baseline methods, as follows:

- Time Series Forecasting Models include Crossformer Zhang & Yan (2023) for multivariate forecasting, TimeMixer Wang et al. (2024b) with multiscale mixing, PatchTST Nie et al. (2022) with channel-independent Transformers, Informer Zhou et al. (2021) for efficient long-sequence forecasting, AutoFormer Wu et al. (2021) with decomposition and autocorrelation, and DLinear Zeng et al. (2023) blending Autoformer and FEDformer with linear layers.
- Graph Neural Network Models include DCRNN Li et al. (2017) integrating diffusion convolution with recurrence, GRUGCN Guan et al. (2024) combining GCN with GRU, EVOLVE-GCN Pareja et al. (2020) for dynamic graphs, and ACGRN Habimana et al. (2020) using attention with convolutional and gated recurrent components.
- Other Deep Learning Models include Transformer Vaswani et al. (2017), Fully Convolutional Recurrent Network (FCRNN) Xie et al. (2016), and Masked Autoencoders (MAE) He et al. (2022).

**Metrics and Other Setting.** Two metrics are used to evaluate the performance of STDACN, i.e., Mean Squared Error(MSE) and Mean Absolute Error(MAE). The smaller the values of MSE and MAE are, the better the prediction effect is. The encoder is composed of 4 layers. The hidden feature dimension is 32 across all formulas (1, 3, 9). And the recursive order of $g^n$Conv in formula

Table 1: **Comparison Experiment on PEMS03 and PEMS04 datasets.**

**Note:** Bold values indicate the best, underlined values are the 2nd best. Values are mean $\pm$ standard deviation, rounded to four decimal places.

| Method | Metric | METR-LA | | | PEMS-BAY | | |
|---|---|---|---|---|---|---|---|
| | | 15 min | 30 min | 1 hour | 15 min | 30 min | 1 hour |
| DCRNN | MSE | 23.9089 ± 0.1527 | 33.2664 ± 0.4808 | 50.3408 ± 0.6579 | 5.7367 ± 0.0347 | 11.0026 ± 0.0616 | 22.2272 ± 1.2995 |
| | MAE | 2.6118 ± 0.0091 | 2.9951 ± 0.0290 | 3.7132 ± 0.0658 | 1.1567 ± 0.0016 | 1.4914 ± 0.0213 | 2.1590 ± 0.1313 |
| Crossformer | MSE | 37.2844 ± 1.9783 | 57.6993 ± 0.8503 | 83.1203 ± 1.6651 | 6.0253 ± 0.1342 | 11.6840 ± 0.2159 | 24.5290 ± 0.3049 |
| | MAE | **2.1574 ± 0.0299** | 3.3667 ± 0.0161 | 3.6924 ± 0.0299 | 1.1615 ± 0.0072 | 1.4794 ± 0.0094 | 2.0486 ± 0.0140 |
| Transformer | MSE | 26.9624 ± 0.0700 | 36.4084 ± 0.0875 | 51.6760 ± 0.1967 | 5.8855 ± 0.0166 | 10.3876 ± 0.0438 | 16.9596 ± 0.0912 |
| | MAE | 2.7647 ± 0.0009 | 3.0443 ± 0.0014 | 3.5266 ± 0.0026 | 1.1640 ± 0.0009 | 1.4240 ± 0.0016 | 1.7626 ± 0.0040 |
| GRUGCN | MSE | 27.3614 ± 0.1289 | 36.7763 ± 0.1335 | 52.0067 ± 0.2243 | 6.0600 ± 0.0124 | 10.7874 ± 0.0226 | 17.9192 ± 0.0677 |
| | MAE | 2.7949 ± 0.0026 | 3.0805 ± 0.0024 | 3.5862 ± 0.0066 | 1.1890 ± 0.0009 | 1.4638 ± 0.0011 | 1.8209 ± 0.0016 |
| EVOLVEGCN | MSE | 28.2634 ± 0.4547 | 38.0367 ± 0.5528 | 54.1686 ± 1.1673 | 6.8048 ± 0.1520 | 11.9731 ± 0.7483 | 18.9596 ± 0.2410 |
| | MAE | 2.9793 ± 0.0258 | 3.3160 ± 0.0349 | 3.8805 ± 0.0428 | 1.2575 ± 0.0036 | 1.5748 ± 0.0269 | 1.9411 ± 0.0118 |
| FCRNN | MSE | 33.5810 ± 0.2417 | 39.4437 ± 0.2245 | 47.7373 ± 0.3385 | 20.4053 ± 0.0860 | 21.6630 ± 0.1237 | 23.2935 ± 0.1411 |
| | MAE | 3.0822 ± 0.0067 | 3.2689 ± 0.0068 | 3.5213 ± 0.0106 | 2.1516 ± 0.0045 | 2.2158 ± 0.0057 | 2.2897 ± 0.0086 |
| TimeMixer | MSE | 57.8965 ± 0.9375 | 101.7485 ± 2.4809 | 152.8255 ± 2.1446 | 5.5164 ± 0.02907 | 13.6072 ± 0.1964 | 24.8927 ± 0.7291 |
| | MAE | 3.1189 ± 0.0638 | 4.1347 ± 0.0212 | 8.7074 ± 0.2841 | 1.1628 ± 0.01496 | 1.6217 ± 0.0182 | 2.1149 ± 0.0253 |
| PatchTST | MSE | 94.3087 ± 2.6543 | 114.4104 ± 3.8901 | 154.1911 ± 2.7341 | 9.9580 ± 0.4760 | 13.4908 ± 0.8306 | 21.1281 ± 0.8042 |
| | MAE | 3.9452 ± 0.1028 | 13.8437 ± 0.7117 | 8.7328 ± 0.2008 | 1.4227 ± 0.0081 | 1.8660 ± 0.0519 | 2.2256 ± 0.9012 |
| Informer | MSE | 86.4470 ± 1.3904 | 139.0305 ± 2.3116 | 371.3622 ± 14.8193 | 10.4633 ± 0.4107 | 18.7256 ± 0.8341 | 181.9524 ± 2.5301 |
| | MAE | 5.0092 ± 0.1012 | 5.9870 ± 0.0391 | 19.8189 ± 1.0594 | 1.8418 ± 0.0735 | 2.3845 ± 0.1042 | 7.5687 ± 0.2491 |
| AutoFormer | MSE | 57.4641 ± 1.2284 | 103.1607 ± 1.7691 | 228.2421 ± 4.2641 | 5.3970 ± 0.0812 | 12.2767 ± 0.4762 | 81.3859 ± 1.6980 |
| | MAE | 3.3332 ± 0.0416 | 4.8013 ± 0.1037 | 14.7905 ± 0.5918 | 1.1785 ± 0.0121 | 1.7677 ± 0.0091 | 3.7912 ± 0.1012 |
| DLinear | MSE | 357.9130 ± 11.4169 | 372.3605 ± 10.2941 | 467.3746 ± 16.6081 | 22.3304 ± 1.8271 | 24.9341 ± 1.6072 | 28.3458 ± 1.7512 |
| | MAE | 7.4557 ± 0.5280 | 9.7525 ± 0.8271 | 13.6325 ± 0.7148 | 2.1570 ± 0.157 | 2.4208 ± 0.0141 | 2.9455 ± 0.0207 |
| ACGRN | MSE | 25.4307 ± 1.5281 | 31.8665 ± 2.3597 | 40.7954 ± 2.8890 | 5.6712 ± 0.2438 | 8.9690 ± 0.6088 | 13.8427 ± 0.8677 |
| | MAE | 2.6416 ± 0.0185 | 2.8467 ± 0.0292 | 3.1526 ± 0.0391 | 1.1505 ± 0.0239 | 1.3627 ± 0.0173 | 1.6569 ± 0.0101 |
| STDACN | MSE | **20.5581 ± 0.0664** | **27.7909 ± 0.2143** | **37.4338 ± 0.4711** | **5.2106 ± 0.0939** | **8.4034 ± 0.0728** | **13.4774 ± 0.1826** |
| | MAE | 2.3817 ± 0.0059 | **2.6570 ± 0.0047** | **3.0522 ± 0.0293** | **1.1276 ± 0.0040** | **1.3389 ± 0.0023** | **1.6284 ± 0.0022** |

(1) is 2. The ratio of the output channels of the down-up type convolution in formula (3) is $r = 4$, and in formula (9) is $r = 2$.

The loss functions of DCRNN, GRU-GCN, EVOLVE-GCN, FCRNN, and ACGRN are Huber loss refer to formula (12), and the batch size are all $64$. Other spatiotemporal series and large-scale model methods adopt the best training parameters. The dataset is split into training, validation, and test sets in an 8:1:1 ratio. The best model after 50 epochs is tested on the test set, averaged over 5 runs.

## 3.2 FORECASTING PERFORMANCE COMPARISON

This subsection presents results across 15-minute, 30-minute, and 1-hour horizons, with the best and 2nd performances bolded and underlined, respectively. The Tables 8 illustrates that STDACN outperformed baseline methods across all test datasets, ranking either first or second in terms of index results. Notably, STDACN outperforms newer models like EVOLVE-GCN, FCRNN, AC-GRN, PatchTST, and TimeMixer due to its multi-layer time recursive gating structure, integrating dynamic convolution kernels and adaptive weight generation to capture temporal dynamics effectively, ideal for non-stationary spatiotemporal sequences. Its dynamic graph convolution module and adaptive spatial calibration parameters enhance dynamic information extraction, surpassing traditional GCN's static limitations, and optimize efficiency, stability, and overall performance through improved temporal-spatial interaction.

## 3.3 HYPERPARAMETER STUDY

This section investigates crucial parameters in the experiment, including temporal recursion level, maximum neighbor link number, and hidden dimension of spatial feature recognition. These parameters play a pivotal role in temporal dimension recognition and spatial feature extraction, influencing the model's innovation level and predictive performance enhancements.

**A. Steps Of Time Recursion Experiment.** The recursive order $K$ of $g^n$Conv from formula (1) is identified as a key hyperparameter affecting gradient updates in the recursive model. The larger $K$ may complicate updates, while the smaller $K$ could hinder efficient temporal interaction, raising the question of its impact. In determining the optimal recursion order $K$, we evaluated model performance on two datasets with input and output lengths ranging from 3 to 12 steps. As shown in Table 2, analysis indicates $K = 2$ yields the highest performance, highlighting that a double-

layer time convolution, as in TCN, boosts the model's ability to capture temporal dynamics and enhance feature extraction for complex patterns in STDACN. However, excessive recursive layers may increase computational overhead and overfitting risks. Thus, two layers strike the optimal balance between accuracy and stability in predictive modeling.

Table 2: **Seps of time recursion Experiment on METR-LA, Solar across 3 to 12 horizons**

| Dataset | Horizon | MSE | | | | MAE | | | |
|---------|---------|-----|-----|-----|-----|-----|-----|-----|-----|
| | Layers | 1 | 2 | 3 | 4 | 1 | 2 | 3 | 4 |
| METR | 3 | 21.0887 | **20.8447** | 21.1404 | 20.9793 | 2.4875 | **2.4735** | 2.4873 | 2.4846 |
| | 4 | 23.5862 | **23.2015** | 23.7405 | 23.5004 | 2.5937 | **2.5805** | 2.6037 | 2.5868 |
| | 5 | 25.8116 | **25.8001** | 26.1318 | 26.4090 | 2.6826 | **2.6760** | 2.6925 | 2.6949 |
| | 6 | 28.1393 | **27.9413** | 28.2526 | 28.6147 | 2.7708 | **2.7276** | 2.7667 | 2.7738 |
| | 7 | **30.1345** | 30.7863 | 30.2364 | 30.8449 | **2.8275** | 2.8589 | 2.8396 | 2.8631 |
| | 8 | 32.4366 | **32.1729** | 32.9769 | 32.5877 | 2.9105 | **2.8980** | 2.9070 | 2.9192 |
| | 9 | 34.2635 | **34.1478** | 34.8145 | 34.1920 | 2.9786 | **2.9684** | 2.9862 | 2.9753 |
| | 10 | 36.0550 | **35.1417** | 35.2847 | 36.5490 | 3.0318 | **3.0242** | 3.0264 | 3.0308 |
| | 11 | 37.1137 | **36.6709** | 37.6002 | 36.8220 | 3.0749 | **3.0265** | 3.1005 | 3.0622 |
| | 12 | 39.0394 | **38.0798** | 39.3730 | 38.8301 | 3.1227 | **3.1056** | 3.1370 | 3.1205 |
| Solar | 3 | 5.6102 | **5.5058** | 5.5758 | 5.5228 | 1.2635 | **1.2256** | 1.2461 | 1.2311 |
| | 4 | 6.9606 | **6.8459** | 6.9303 | 6.9468 | 1.4329 | **1.4162** | 1.4439 | 1.4406 |
| | 5 | 8.3877 | **8.2283** | 8.2853 | 8.4649 | 1.6185 | **1.6068** | 1.6145 | 1.6309 |
| | 6 | 9.8233 | **9.4471** | 9.6664 | 9.6812 | 1.7792 | **1.7402** | 1.7787 | 1.7714 |
| | 7 | 10.9045 | **10.8692** | 10.9611 | 10.9843 | **1.8998** | 1.9026 | 1.9198 | 1.9092 |
| | 8 | **12.3331** | 12.3589 | 12.4400 | 12.4110 | 2.0532 | **2.0458** | 2.0484 | 2.0594 |
| | 9 | 13.8980 | **13.7772** | 13.8023 | 13.7815 | 2.1820 | **2.1676** | 2.2020 | 2.1925 |
| | 10 | 14.9697 | **14.9048** | 15.3873 | 15.1245 | 2.3037 | **2.2293** | 2.3429 | 2.3118 |
| | 11 | **16.2645** | 16.5844 | 16.8956 | 16.6476 | 2.4266 | 2.4191 | **2.4119** | 2.4422 |
| | 12 | 18.5231 | **17.7397** | 17.9982 | 18.2235 | 2.6193 | **2.5358** | 2.5859 | 2.5859 |

**B. Maximum Neighborhood Connections.** The study evaluates the model's performance with varying Maximum Neighborhood Connections on two datasets over 3, 6, and 12-month spans. This parameter controls the number of neighboring nodes in the adaptive adjacency matrix, balancing computational efficiency and node interconnection capture for graph sparsification. Table 3 shows that 128 connections typically optimize performance, accuracy, and stability. The slight performance variation highlights the model's adaptability, enhancing its predictive capabilities across diverse scenarios.

Table 3: **Max. neighborhood connect steps Experiment**

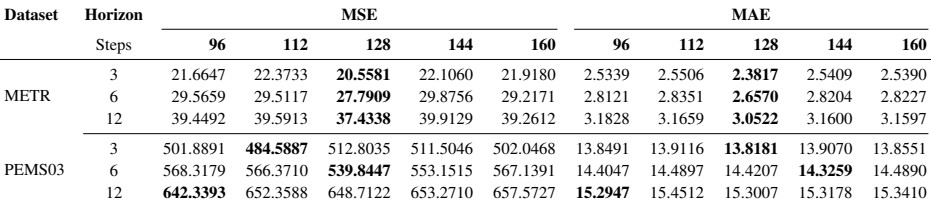

| Dataset | Horizon | MSE | | | | | MAE | | | | |
|---------|---------|-----|-----|-----|-----|-----|-----|-----|-----|-----|-----|
| | Steps | 96 | 112 | 128 | 144 | 160 | 96 | 112 | 128 | 144 | 160 |
| METR | 3 | 21.6647 | 22.3733 | **20.5581** | 22.1060 | 21.9180 | 2.5339 | 2.5506 | **2.3817** | 2.5409 | 2.5390 |
| | 6 | 29.5659 | 29.5117 | **27.7909** | 29.8756 | 29.2171 | 2.8121 | 2.8351 | **2.6570** | 2.8204 | 2.8227 |
| | 12 | 39.4492 | 39.5913 | **37.4338** | 39.9129 | 39.2612 | 3.1828 | 3.1659 | **3.0522** | 3.1600 | 3.1597 |
| PEMS03 | 3 | 501.8891 | **484.5887** | 512.8035 | 511.5046 | 502.0468 | 13.8491 | 13.9116 | **13.8181** | 13.9070 | 13.8551 |
| | 6 | 568.3179 | 566.3710 | **539.8447** | 553.1515 | 567.1391 | 14.4047 | 14.4897 | 14.4207 | **14.3259** | 14.4890 |
| | 12 | **642.3393** | 652.3588 | 648.7122 | 653.2710 | 657.5727 | **15.2947** | 15.4512 | 15.3007 | 15.3178 | 15.3410 |

**C. Embedding Dimensions Experiment.** This section evaluates the model's predictive performance on two datasets using spatiotemporal feature embedding dimensions of 6, 8, 10, and 12. Table 4 shows optimal performance at a specific dimension, highlighting its superior predictive ability. Adjusting embedding dimensions to dataset characteristics improves accuracy, emphasizing the importance of optimization for enhanced prediction and adaptability.

Table 4: **Embedding Dimensions Experiment**

| Dataset | Horizon | MSE | | | | MAE | | | |
|---------|---------|-----|-----|-----|-----|-----|-----|-----|-----|
| | Embed Dim | 6 | 8 | 10 | 12 | 6 | 8 | 10 | 12 |
| METR | 3 | 21.9693 | 21.5281 | **20.5581** | 21.6883 | 2.5485 | 2.5369 | **2.3817** | 2.5390 |
| | 6 | 39.1484 | 29.3826 | **27.7909** | 29.4906 | 2.8079 | 2.8143 | **2.6570** | 2.8238 |
| | 12 | 39.1484 | 39.9738 | **37.4338** | 39.5605 | 3.1551 | 3.1942 | **3.0522** | 3.1735 |
| PEMS03 | 3 | 536.7852 | 517.5540 | **512.8035** | 503.7646 | 13.9394 | 13.8256 | **13.8181** | 13.8957 |
| | 6 | 580.0443 | 570.4965 | **539.8447** | 553.5356 | 14.4459 | 14.4269 | **14.4207** | 14.4772 |
| | 12 | 649.3734 | 658.3918 | 648.7122 | **620.2323** | 15.3388 | 15.2591 | **15.2007** | 15.2849 |

## 3.4 COMPONENT EXPERIMENTS

The component experiment results are shown in Table 5, which examine model performance across various activation functions, convolution types, and parameters to find the best settings. Activation functions like delta, Sigmoid, and Tanh underperform compared to the Mish function, which offers clear advantages. We also find that higher-order gated convolution designs outperform 2D Conv, gated Conv, and gated + 2D Conv. Additionally, adaptive dynamic graph convolution surpasses traditional GCN variants (original, adaptive, and dynamic GCN). These results highlight STDACN's unique component integration, boosting prediction accuracy and robustness for complex ST-data.

Table 5: **Different components performance Experiment**

| Method Component | MSE | | | MAE | | |
|---|---|---|---|---|---|---|
| | 3 | 6 | 12 | 3 | 6 | 12 |
| **Activation Function Comparison** | | | | | | |
| STDACN | **20.5582** | **27.7909** | **37.4339** | **2.3817** | **2.6570** | **3.0522** |
| $\delta$ Activation | 25.9292 | 33.4192 | 43.6290 | 2.7994 | 3.0349 | 3.3773 |
| Sigmoid Activation | 22.1816 | 29.7810 | 39.5736 | 2.5651 | 2.8352 | 3.1964 |
| Tanh Activation | 21.6930 | 29.5346 | 40.0122 | 2.5292 | 2.8159 | 3.1628 |
| **Convolution Type Comparison** | | | | | | |
| STDACN | **20.5582** | **27.7909** | **37.4339** | **2.3817** | **2.6570** | **3.0522** |
| 2D Conv. | 24.7999 | 33.5339 | 45.5572 | 2.7007 | 2.9839 | 3.4138 |
| Gated Conv. | 22.1050 | 29.1382 | 39.5585 | 2.5528 | 2.8185 | 3.1784 |
| Gated+2D Conv. | 22.5234 | 29.6166 | 39.1621 | 2.5923 | 2.8435 | 3.1919 |
| **Graph Convolution Comparison** | | | | | | |
| STDACN | **20.5582** | **27.7909** | **37.4339** | **2.3817** | **2.6570** | **3.0522** |
| GCN | 21.3318 | 28.0773 | 37.5429 | 2.5123 | 2.7439 | 3.0766 |
| Adapt. GCN | 21.7875 | 29.8753 | 38.1135 | 2.5558 | 2.8367 | 3.1333 |
| Adapt. w/o $\beta$ | 22.1189 | 29.2167 | 39.8782 | 2.5472 | 2.8098 | 3.1651 |

## 3.5 ABLATION EXPERIMENTS

The ablation study assessed the impact of removing key modules—spatial self-learning matrix, temporal causal convolution, spatial adaptive module, and dynamic learning coefficient—on performance, as shown in Fig. 3 for METR-LA, PEMS-BAY, and Solar datasets across 3, 6, and 12-step horizons. The spatial self-learning matrix updates topology dynamically, enhancing resolution, while adaptive temporal causal convolution outperforms fixed-kernel TCNs. The spatial adaptive module and dynamic learning coefficients in GCNs derive calibration weights $\beta$ from $X_{dt}^{(l)}$, improving spatial dependency modeling. Results confirm each component's critical role, especially spatial adaptation and self-learning matrices, in enhancing ST-prediction accuracy across diverse datasets.

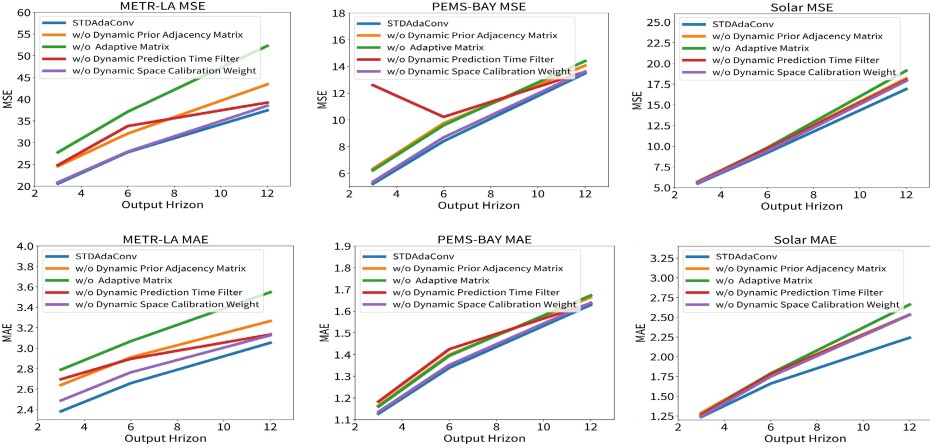

Figure 3: Ablation Experiments.

## 3.6 EFFICIENCY EXPERIMENT

Efficiency Analysis Experiment We analyze the differences between STDACN model and other mainstream spatiotemporal data prediction methods in terms of model size, training time, inference time, and model accuracy. The results presented in Table 6, the method maintains the second largest model size, while its training time and inference time are also short, only lagging behind the lighter model, but providing the best prediction accuracy. These results prove that the model has high efficiency and optimal performance for large-scale spatiotemporal tasks, and is excellent in reasoning speed and analysis accuracy.

Table 6: **Efficiency comparison of various methods.**

| Methods | In/Out | Model size | Training time | Inference Time | MSE | MAE |
|---|---|---|---|---|---|---|
| TimeMixer | | 175,177 | 26.1935 | 62.3523 | 152.8255 | 8.7074 |
| Crossformer | | 2,335,116 | 178.9818 | 6.5167 | 83.1203 | 3.6924 |
| PatchTST | | 113,579 | **9.5499** | 3.5398 | 154.1911 | 8.7328 |
| Informer | Input:12 | 7,322,831 | 22.3132 | 0.6114 | 371.3622 | 19.8189 |
| AutoFormer | Output:12 | 6,842,575 | 24.8419 | 0.0311 | 228.2421 | 14.7905 |
| DLinear | | **2,328** | 15.5330 | **0.0016** | 467.3746 | 13.6325 |
| Transformer | | 6,534,863 | 16.8897 | 0.7171 | 51.676 | 3.5266 |
| ACGRN | | 751,650 | 550.3586 | 0.0286 | 40.7954 | 3.1526 |
| STDACN | | 87,858 | 15.3111 | 0.5418 | **37.4339** | **3.0522** |

## 3.7 ANTI-NOISE EXPERIMENT

This section validates STDACN's performance under spatial noise, with Table 7 showing results for 3, 6, and 12-step predictions on METR-LA, PEMS03, and Solar datasets, comparing normal data to 20%, 60%, and 100% noise levels. METR-LA exhibits minimal error increase, PEMS03 shows no significant degradation, while Solar data is more noise-sensitive in long-term predictions. STDACN maintains stability with less than 5% performance loss, demonstrating the dynamic adaptive spatial module's effectiveness in handling noise for reliable real-world predictions.

Table 7: **Anti-noise Analysis Capability Examination**

| Dataset | Condition | MSE | | | MAE | | |
|---|---|---|---|---|---|---|---|
| | | 3 | 6 | 12 | 3 | 6 | 12 |
| METR-LA | 100% Data | **20.5581** | **27.7909** | **37.4338** | **2.3817** | **2.6570** | **3.0522** |
| | 20% Noise | 21.6851 | 28.9091 | 40.1571 | 2.5320 | 2.8005 | 3.1979 |
| | 60% Noise | 22.3113 | 29.4559 | 40.0812 | 2.5537 | 2.8213 | 3.1971 |
| | 100% Noise | 21.7943 | 28.9002 | 40.0186 | 2.5342 | 2.8089 | 3.2183 |
| PEMS03 | 100% Data | 512.8035 | **539.8447** | 648.7122 | 13.8181 | 14.4207 | 15.3007 |
| | 20% Noise | 522.8550 | 572.2609 | 687.9156 | 13.8074 | 14.4293 | 15.5739 |
| | 60% Noise | **477.7003** | 586.0765 | 692.9213 | 13.7968 | 14.5882 | 15.9085 |
| | 100% Noise | 513.0293 | 558.4977 | **647.3211** | **13.7266** | **14.3808** | **15.2827** |
| Solar | 100% Data | 5.4933 | **9.2019** | **16.9010** | 1.2369 | **1.6599** | **2.2409** |
| | 20% Noise | 5.6141 | 9.4426 | 25.4844 | 1.2556 | 1.7246 | 3.1514 |
| | 60% Noise | 5.5416 | 9.2854 | 26.7097 | 1.2518 | 1.7133 | 3.2582 |
| | 100% Noise | **5.2492** | 9.5612 | 26.3236 | **1.2005** | 1.7448 | 3.1987 |

## 4 CONCLUSION AND FUTURE

The STDACN framework uses dynamic and adaptive convolutional networks, including a high-order gated Temporal Convolutional Network (TCN) and an adaptive dynamic Graph Convolutional Network (GCN), to effectively capture complex spatiotemporal dependencies, boosting prediction capabilities. Its optimized hyperparameters and robust performance under noisy conditions highlight its potential for real-world use. With a streamlined parameter count and efficient training and inference times, STDACN is highly practical across domains. Future work could integrate multimodal data (e.g., weather, social media, traffic sensors), extend frameworks for real-time edge computing, enhance scalability for large networks, and explore advanced loss functions or attention mechanisms to improve robustness and adaptability while addressing computational constraints and evolving data patterns.

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

# A    APPENDIX

## A.1    RELATED WORK

**Spatiotemporal prediction method based on shared convolution kernel.** As shown in Fig. 1, spatiotemporal data feature tensors include temporal, spatial, and feature dimensions, with accurate prediction depending on effectively characterizing their relationships. Recent research mainly focuses on statistically analyzing inherent data relationships. For instance, HDL-net Bao et al. (2019) employs a multilayer ConvLSTM for capturing temporal and spatial characteristics of shared bicycle demand, TSTGCNZhang et al. (2021) integrates CNN and GCN for spatiotemporal attention prediction, ST-HSL Li et al. (2022b) combines CNN and TCN for time series analysis, and incorporates spatial dependence through hypergraph information maximization. Nevertheless, most methods rely on shared convolution kernels for space-time analysis, hindering the extraction of dynamic patterns varying with time and space.

**Spatiotemporal prediction methods based on static graph structure.** Many spatial relation modeling methods rely on GCNKipf & Welling (2016). These methods are classified into explicit (EX-GCN) and implicit (IM-GCN) Wang et al. (2022) based on spatial topological relation construction, with EX-GCN using physical relations and IM-GCN leveraging semantic-derived implicit relations, sparking growing research interest in IM-GCN. SLCNN Zhang et al. (2020) incorporates global and local local SLC module relationships for predicting traffic spatiotemporal data. AGCRN Bai et al. (2020) establishes hidden spatial relationships through node attribute learning, while Graph Wavenet Wu et al. (2019) employs an adaptive approach to learn global spatial relationships. DGCRN Li et al. (2021) dynamically generates implicit spatial relations to learn dynamic topological relations. However, these approaches often overlook the interconnectedness between global and local, or static and dynamic spatial topological relationships, which are crucial in practical applications.

**Spatiotemporal prediction methods based on large-scale models.** Integrating spatiotemporal feature learning with transformer architecture boosts prediction accuracy and robustness but faces computational efficiency and power demand issues. SimMTM Dong et al. (2023) enhances time series prediction and classification with masking and manifold learning, though multipoint aggregation reduces efficiency. AdaMAE Bandara et al. (2023) uses adaptive masking and reinforcement learning for video classification, but iterative optimization increases costs. TimeGPT Garza et al. (2023) excels in homodyne inference yet struggles with real-time use due to complexity. ST-LLM Liu et al. (2024) employs high-parameter spatiotemporal embedding for dynamic system modeling. Despite improved accuracy, these methods face resource constraints, efficiency challenges in large-scale scenarios, and latency in real-time applications.

**Dynamic spatiotemporal prediction methods.** The local connectivity inherent Huang et al. (2021) in static convolution yields translation invariance, whereas dynamic convolution enhances the performance of existing convolutional models by generating new weight parameters through the dynamic integration of multiple convolution kernels. For instance, CondConvYang et al. (2019) introduces a conditional parameter convolution approach that assigns a specific convolution kernel parameter to each example, thereby increasing model size and capacity without compromising computational efficiency. Building upon this concept, ODConv Li et al. (2022a) extends the one-dimensional dynamic properties of CondConv to incorporate spatial, input, and output channel dynamics. TAdaConv Huang et al. (2021) introduces a temporal adaptive convolution algorithm tailored for video comprehension. However, this multi-convolution dynamic optimization mechanism will greatly increase the operation cost and affect the efficiency and generalization ability of the model.

## A.2    MATHEMATICAL DEFINITION

This paper aims to enhance spatiotemporal prediction accuracy using dynamic spatiotemporal convolution.

**Definition 1** *We use the spatial topological relation network as a weighted undirected graph $G = (V, E, A)$ to describe the structure of the space relationship, where $V = \{v_0, \cdots, v_N\}$ is $N$ spatial*

nodes, the adjacency matrix $A \in \mathbb{R}^{N \times N}$ is used to represent the connection strength. The $G$ is dynamically changing with time, which is recorded as $G_t$.

**Definition 2** *Spatiotemporal feature matrix $X$. The information on the spatial relationship network $G$ is regarded as attribute features of nodes $V$, which is indicated by $X \in \mathbb{R}^{F \times N \times T}$, where $F$ is the number of node attribute features, $T$ represents the length of the historical time series, and $N$ is the number of sensor nodes.*

The problem of spatiotemporal prediction is considered to predict future data $\hat{Y} = (\hat{x}_{T+1}, \cdots, \hat{x}_{T+\tau})$ from current data $X = (x_1, \cdots, x_T)$. With the above definition, we should learn the mapping function $f$ from $X$ to $\hat{Y}$, that is:

$$\hat{Y} = f_\theta(X, G), \tag{13}$$

where $\theta$ is the model parameter. the real future data are $Y = (x_{T+1}, \cdots, x_{T+\tau})$ and the training process makes the distance between $\hat{Y}$ and $Y$ increasingly smaller.

### A.3 SUPPLEMENTARY EXPERIMENTS

We conducted additional comparative experiments on the dataset and carried out a more comprehensive performance analysis. Our model continues to demonstrate superior performance; the results are as follows:

Table 8: **Comparison Experiment on PEMS03 and PEMS04 datasets.**

| Method | Metric | PEMS03 | | | PEMS08 | | |
|---|---|---|---|---|---|---|---|
| | | 15 min | 30 min | 1 hour | 15 min | 30 min | 1 hour |
| DCRNN | MSE | 572.9481 ± 6.2247 | 674.2108 ± 5.3580 | 846.0167 ± 13.3477 | 538.2844 ± 0.8367 | 666.7077 ± 10.9641 | 1386.2029 ± 147.4780 |
| | MAE | 14.2457 ± 0.0431 | 15.3870 ± 0.0820 | 17.5119 ± 0.1082 | 15.1848 ± 0.0504 | 17.0471 ± 0.1884 | 25.4170 ± 1.7328 |
| CFORMER | MSE | **345.4315 ± 6.1329** | 569.8165 ± 17.9203 | **499.9477 ± 26.5685** | 750.9642 ± 4.2122 | 863.7393 ± 11.4401 | 1092.6679 ± 12.8182 |
| | MAE | **12.6388 ± 0.0869** | 15.4411 ± 0.2115 | **14.7761 ± 0.2656** | 19.5578 ± 0.0338 | 20.7773 ± 0.1361 | 21.2692 ± 0.0795 |
| Transformer | MSE | 557.4231 ± 20.9859 | 615.8816 ± 16.6908 | 725.2109 ± 6.6085 | 538.9656 ± 0.8856 | 598.6457 ± 2.3719 | 693.0868 ± 4.1460 |
| | MAE | 13.8957 ± 0.0565 | 14.5521 ± 0.0406 | 15.8781 ± 0.0318 | 14.7976 ± 0.0416 | 15.3489 ± 0.0463 | 16.3003 ± 0.0616 |
| GRUGCN | MSE | 526.1056 ± 1.3160 | 630.4568 ± 1.7699 | 805.1929 ± 5.1585 | 577.5853 ± 1.4448 | 690.5662 ± 1.4979 | 880.8742 ± 2.9947 |
| | MAE | 14.5308 ± 0.0132 | 15.6593 ± 0.0244 | 17.5197 ± 0.0364 | 15.4032 ± 0.0207 | 16.7139 ± 0.0215 | 18.7708 ± 0.0264 |
| EVOLVEGCN | MSE | 659.6220 ± 86.3013 | 878.9086 ± 50.3883 | 978.7947 ± 49.7286 | 679.0293 ± 3.3324 | 802.6686 ± 4.3030 | 5141.8217 ± 8253.8358 |
| | MAE | 15.8614 ± 0.0620 | 17.1105 ± 0.0391 | 18.9810 ± 0.0662 | 16.7804 ± 0.0334 | 18.1757 ± 0.0440 | 40.5428 ± 40.1884 |
| FCRNN | MSE | 1117.6096 ± 23.8709 | 1131.6569 ± 28.2207 | 1143.9044 ± 8.2393 | 1167.0077 ± 14.1085 | 1183.4153 ± 27.9255 | 1212.4311 ± 10.1895 |
| | MAE | 18.8406 ± 0.1791 | 19.1202 ± 0.2217 | 19.3827 ± 0.0730 | 20.7240 ± 0.1020 | 21.0194 ± 0.1594 | 21.3018 ± 0.0615 |
| TimeMixer | MSE | 580.6508 ± 22.2139 | 686.3181 ± 36.1629 | 1073.0527 ± 70.3147 | 598.3445 ± 21.1839 | 745.6439 ± 19.1338 | 1088.7059 ± 37.4260 |
| | MAE | 14.5278 ± 0.8647 | 16.9218 ± 1.1092 | 20.5205 ± 1.2733 | 15.8164 ± 1.2401 | 17.4957 ± 1.1637 | 20.8164 ± 1.1143 |
| PatchTST | MSE | 576.3602 ± 22.6158 | 614.0498 ± 30.8192 | 691.7393 ± 24.4209 | 541.7598 ± 18.3164 | 617.3783 ± 23.1965 | 764.9968 ± 29.3392 |
| | MAE | 13.8834 ± 0.8174 | **14.3375 ± 0.9527** | 16.7915 ± 0.9819 | 15.1952 ± 1.0334 | 16.8763 ± 0.9171 | 17.7116 ± 0.9842 |
| Informer | MSE | 856.0954 ± 45.2617 | 975.9937 ± 39.1326 | 1222.3432 ± 108.8122 | 582.4223 ± 28.4113 | 639.0414 ± 31.2371 | 667.5425 ± 22.5188 |
| | MAE | 16.4171 ± 0.9177 | 17.8803 ± 1.1283 | 19.7738 ± 1.2507 | 15.0627 ± 0.8390 | 15.8433 ± 0.8274 | 16.1083 ± 0.7486 |
| AutoFormer | MSE | 522.4533 ± 17.5178 | 552.8334 ± 22.0388 | 1878.6222 ± 187.2339 | 685.2832 ± 27.1372 | 727.3796 ± 31.5008 | 813.4028 ± 33.1468 |
| | MAE | 13.9836 ± 0.6496 | 15.1984 ± 0.8172 | 28.8514 ± 1.3760 | 16.3894 ± 1.0334 | 17.1013 ± 0.9012 | 18.5997 ± 1.1331 |
| DLinear | MSE | 9299.81 ± 437.52 | 9802.14 ± 347.21 | 11782.53 ± 898.10 | 3930.44 ± 198.72 | 4583.53 ± 274.51 | 5206.44 ± 308.92 |
| | MAE | 63.5546 ± 2.9833 | 65.5238 ± 2.8972 | 76.7675 ± 5.9350 | 26.1485 ± 1.4807 | 28.9994 ± 1.8211 | 30.7945 ± 2.1609 |
| ACGRN | MSE | 46317.02 ± 1807.21 | 62880.80 ± 2471.41 | 52807.63 ± 1847.10 | 119761.65 ± 2398.51 | 125888.52 ± 3182.92 | 120153.48 ± 3183.77 |
| | MAE | 168.9331 ± 8.1548 | 203.1996 ± 13.1372 | 186.2947 ± 17.2339 | 317.3416 ± 21.6239 | 322.2049 ± 28.7401 | 303.3234 ± 19.0326 |
| STDACN | MSE | 512.8035 ± 15.4352 | **539.8447 ± 12.6211** | 648.7122 ± 18.4802 | 528.9281 ± 2.8034 | **587.4261 ± 3.7328** | **654.0064 ± 4.4649** |
| | MAE | 13.8181 ± 0.0315 | 14.4207 ± 0.0586 | 15.3007 ± 0.0529 | **14.7299 ± 0.0442** | **15.3360 ± 0.0375** | **16.2167 ± 0.0723** |

