# OpenReview forum: "STDACN: a Spatiotemporal Prediction Framework based on Dynamic and Adaptive Convolution Networks"
_ICLR.cc/2026/Conference — ICLR 2026 Conference Withdrawn Submission_

### Official Review · Reviewer_JMXN · 2025-10-26

**Soundness:** 2
**Presentation:** 2
**Contribution:** 2
**Rating:** 2
**Confidence:** 4

**Summary:**

This paper proposes STDACN, a spatiotemporal prediction framework combining high-order gated TCN with recursive causality and adaptive GCN using diffusion convolution.

**Strengths:**

- Addresses a relevant problem in spatiotemporal modeling
- Combines temporal and spatial modules in a unified framework
- Evaluation on real-world datasets (traffic, electricity)
- The general motivation for moving beyond strict weight sharing is reasonable

**Weaknesses:**

- Dynamic convolution via channel calibration (Eq. 2-5) resembles existing attention mechanisms like SENet and CondConv. The recursive gating appears as a minor modification of standard gated TCN. The distinction between this work and prior self-learning graph methods is unclear.

- The GCN formulation (Eq. 6) is incomplete in the submission. Missing details on how temporal dynamics integrate with spatial adaptive learning. The "dynamic diffusion convolution" mechanism needs fuller explanation.

- Experimental validation is insufficient with only two dataset types mentioned. No ablation studies demonstrate the contribution of individual components. Computational cost analysis is absent. The reported 1.2%-4.7% improvement range is ambiguous.

- Theoretical analysis is lacking. No justification for why recursive gating helps or under what conditions. The choice of mish activation for gradient issues needs support. Hyperparameter selection (e.g., rt in Eq. 4) appears arbitrary.

**Questions:**

1. How does your dynamic calibration mechanism differ from squeeze-and-excitation or dynamic convolution networks? Can you provide direct comparisons?

2. What is the computational overhead of the K-order recursive gating? How do you choose K?

3. The claim about "breaking weight sharing constraints"—can you formalize what constraint is broken and prove this improves expressiveness?

4. Can you provide ablation studies showing: (a) recursive gating vs. standard gating, (b) mish vs. tanh, (c) dynamic calibration vs. static kernels?

5. How does performance scale with graph size? What about computational complexity?

---

### Official Review · Reviewer_GLJq · 2025-10-29

**Soundness:** 2
**Presentation:** 2
**Contribution:** 2
**Rating:** 4
**Confidence:** 3

**Summary:**

This paper introduces **STDACN**, a spatio-temporal forecasting model designed to overcome the limitations of **static weight-sharing** in conventional convolutional networks. The architecture is hierarchical, combining three main dynamic components: a **recursive high-order gated TCN** ($g^n$Conv), a **Dynamic Causal Temporal Convolution (DCTC)** that generates channel-wise calibration weights $\pi$, and an **Adaptive Dynamic GCN (SAGC)** that learns a time-varying adjacency matrix $\tilde{A}_{\text{adp}}$. Experiments across five datasets (METR-LA, PEMS-BAY, PEMS03/08, and Solar) demonstrate competitive performance and improved robustness compared to a broad range of established baselines.

**Strengths:**

1. **Motivation and Coherent Design.** STDACN directly addresses static weight-sharing/stationarity limitations in TCNs and GCNs, integrating temporal adaptivity (DCTC) and spatial adaptivity (SAGC) into a unified framework.
2. **Competitive Empirical Performance.** The model frequently ranks top-1 or top-2 across prediction horizons and diverse datasets compared to many baselines (GNN, TCN, Transformer families).
3. **Thorough Component Analysis.** Ablation studies, convolution type comparisons, and graph variant analyses clearly show the contributions of individual design choices.
4. **Demonstrated Robustness and Efficiency.** Anti-noise experiments and efficiency analysis (Table 6) support practical applicability with favorable tradeoffs relative to larger baselines.

**Weaknesses:**

1. **Limited Technical Novelty.** The architecture integrates existing concepts (dynamic filters, channel-wise attention, adaptive GCNs) without isolating clear novel contributions or justifying the choice of components.
2. **Ambiguity in Dynamic Mechanisms.** The operations for temporal calibration ($\pi$) and spatial coefficients ($\beta$) are unclear; it is not specified whether $\pi$ is per-channel, full kernel generator, or feature modulator.
3. **Inconsistent Notation & Reproducibility Gaps.** Key dimensions (e.g., $K$, $P$, $r$) are inconsistently defined, and details on optimizer, learning rate schedule, weight decay, and dropout are missing.
4. **Insufficient Baseline Tuning and Statistical Validation.** The hyperparameter search for baselines is not detailed, and no statistical significance tests confirm that improvements are reliable.
5. **Lack of Qualitative Interpretation.** No visualizations are provided for temporal evolution of $\pi(t)$ or adaptive adjacency $\tilde{A}_{\text{adp}}(t)$, which would help verify meaningful learning of dynamic components.

**Questions:**

1. **Calibration Mechanism.** How is the calibration vector $\pi=\Pi(X)$ applied to the temporal kernels $W$? Please provide tensor shapes and a clear forward-pass equation for the DCTC block.
2. **Adaptive Adjacency.** What is the exact formula for $\tilde{A}_{\text{adp}}$? How is it normalized, and is it computed per time step or via learned static parameters?
3. **Regularization & Overfitting.** Which strategies (weight decay, dropout) are used to prevent overfitting given the added flexibility from dynamic kernels?
4. **Visualization Request.** Please provide plots of predicted vs. ground-truth traces, and visualizations of learned dynamic components, e.g., $\pi(t)$ and $\tilde{A}_{\text{adp}}(t)$, over periods including significant events (e.g., rush hour).
5. **Baselines and Tuning.** Describe the hyperparameter search protocol for baselines and confirm that each received comparable compute budget for fair comparison.

---

### Official Review · Reviewer_c7zK · 2025-10-30

**Soundness:** 2
**Presentation:** 1
**Contribution:** 1
**Rating:** 2
**Confidence:** 4

**Summary:**

This paper proposes STDACN, a model for spatio-temporal prediction, which aims to improve upon static TCNs and GCNs. The method consists of two main components: (1) a high-order dynamic gated Temporal Convolutional Network (TCN) and (2) an adaptive dynamic Graph Convolutional Network (GCN).

**Strengths:**

S1 - The paper proposes a lightweight model that shows comparable empirical results on traffic forecasting benchmarks when compared to significantly larger Transformer-based models.

**Weaknesses:**

W1 - The paper suffers from a significant lack of novelty. The core concepts of dynamic gating convolutions and adaptive graph convolutions are well-established. The paper fails to position its contribution relative to prior methods adequately. It composes a pipeline of prior techniques rather than a new, principled approach.

W2 - The empirical evaluation is flawed and insufficient to support the paper's claims. The paper omits comparisons to several state-of-the-art dynamic graph models that are directly relevant to this work. These omissions include DGCRN [1], a key work in dynamic graph-based forecasting, as well as more recent and high-performing methods like MSTFGRN [2] and SDSINet [3]

[1] Luo, Xunlian, et al. "Dynamic graph convolutional network with attention fusion for traffic flow prediction." arXiv preprint arXiv:2302.12598 (2023).

[2] Zhao, Wei, et al. "Multi-spatio-temporal fusion graph recurrent network for traffic forecasting." Engineering Applications of Artificial Intelligence 124 (2023): 106615.

[3] Yang, Shiyu, and Qunyong Wu. "SDSINet: A spatiotemporal dual-scale interaction network for traffic prediction." Applied Soft Computing 173 (2025): 112892.

**Questions:**

Could you please confirm and correct the presentation errors?

Q1 - Is Table 1 for METR-LA and PEMS-BAY?

Q2 - Is Table 8 for PEMS03 and PEMS08?

Q3 - Can authors provide details of Solar dataset?

---

### Official Review · Reviewer_j3gX · 2025-10-31

**Soundness:** 2
**Presentation:** 2
**Contribution:** 2
**Rating:** 4
**Confidence:** 4

**Summary:**

The paper proposes a Spatiotemporal Dynamic Adaptive Convolution Network (STDACN) for forecasting complex spatiotemporal data such as traffic and energy flow. It integrates a high-order dynamic gated TCN to capture long-term temporal dependencies and an adaptive dynamic GCN to model time-varying spatial relationships. Dynamic calibration mechanisms allow the model to adjust convolutions and adjacency matrices adaptively.

**Strengths:**

The paper presents a clear and technically sound framework that combines TCN and GCN with dynamic adaptive mechanisms. The integration of dynamic gating and adaptive graph convolution improves the model’s ability to capture time-varying spatiotemporal dependencies. Overall, the approach is well-motivated and contributes to advancing dynamic deep learning methods for spatiotemporal prediction.

**Weaknesses:**

1. The paper’s originality appears limited. The proposed dynamic temporal module closely resembles the TCN component in Graph WaveNet [1], with only minor modifications such as replacing the activation with Mish and introducing input-driven dynamic weight adjustment. Similarly, the dynamic GCN shares strong conceptual overlap with the adaptive adjacency mechanisms in Graph WaveNet, with limited methodological novelty.
2. Notation and formulation inconsistencies also affect clarity. The motivation for up/down-sampling and the use of the hyperparameter r_t are not clearly justified, and their effects are not analyzed experimentally.
3. Dataset usage and experimental reporting are incomplete: the Solar dataset is not introduced, while datasets mentioned in the abstract and introduce (e.g., population, electricity) do not appear in the experiments.
4. Baseline selection omits several strong and widely recognized spatiotemporal forecasting models (e.g., Graph WaveNet [1], PDFormer [2], HimNet [3], STD-PLM [4]).

[1] Wu Z, Pan S, Long G, et al. Graph wavenet for deep spatial-temporal graph modeling[J]. arXiv preprint arXiv:1906.00121, 2019.

[2] Jiang J, Han C, Zhao W X, et al. Pdformer: Propagation delay-aware dynamic long-range transformer for traffic flow prediction[C]//Proceedings of the AAAI conference on artificial intelligence. 2023, 37(4): 4365-4373.

[3] Dong Z, Jiang R, Gao H, et al. Heterogeneity-informed meta-parameter learning for spatiotemporal time series forecasting[C]//Proceedings of the 30th ACM SIGKDD conference on knowledge discovery and data mining. 2024: 631-641.

[4] Huang Y, Mao X, Guo S, et al. Std-plm: Understanding both spatial and temporal properties of spatial-temporal data with plm[C]//Proceedings of the AAAI Conference on Artificial Intelligence. 2025, 39(11): 11817-11825.

**Questions:**

1. Could the authors clarify the definition of P in X \in \mathbb{R}^{F\times N\times P} and the dual use of k for both kernel size and recursion depth?
2. The Diffusion Convolution seems conceptually part of the GCN but appears in the TCN block in Figure 2 — could the authors clarify this architectural inconsistency?
3. What is the rationale for the up/down-sampling design in the TCN block, and how does the hyper-parameter r_t influence temporal aggregation? Why is it not analyzed in the hyper-parameter study?
4. Is the FC layer intended to map H to C_{in}? If so, why is \pi not shaped as C_{in} \times 1 \times \frac{P(l)}{r_t}?
5. Why is the Solar dataset omitted from the dataset description, and what are the “population” and “electricity” datasets mentioned in the abstract and introduce?
6. Could the authors justify the baseline choices and clarify whether comparisons were made against leading spatiotemporal models (e.g., GWNet, PDFormer, HimNet, STD-PLM)?
7. On which dataset were the component experiments conducted?
8. Please check the consistency of table titles and correct typographical errors across the paper.

---

### Note · Authors · 2025-11-30

I have read and agree with the venue's withdrawal policy on behalf of myself and my co-authors.